# Histopathological Spectrum and Molecular Characterization of Liver Tumors in the Setting of Fontan-Associated Liver Disease

**DOI:** 10.3390/cancers16020307

**Published:** 2024-01-11

**Authors:** Paola Francalanci, Isabella Giovannoni, Chantal Tancredi, Maria Giulia Gagliardi, Rosalinda Palmieri, Gianluca Brancaccio, Marco Spada, Giuseppe Maggiore, Andrea Pietrobattista, Lidia Monti, Aurora Castellano, Maria Cristina Giustiniani, Andrea Onetti Muda, Rita Alaggio

**Affiliations:** 1O.U. Pathology, Bambino Gesù Children’s Hospital, IRCCS, Piazza Sant’Onofrio 4, 00165 Rome, Italy; isabella.giovannoni@opbg.net (I.G.); chantal.tancredi@opbg.net (C.T.); andrea.onettimuda@opbg.net (A.O.M.); rita.alaggio@opbg.net (R.A.); 2DPCCS Adult Congenital Cardiology, Bambino Gesù Children’s Hospital, IRCCS, 00165 Rome, Italy; mgiulia.gagliardi@opbg.net (M.G.G.); rosalinda.palmieri@opbg.net (R.P.); gianluca.brancaccio@opbg.net (G.B.); 3Hepatobiliary and Transplant Surgery, Bambino Gesù Children’s Hospital, IRCCS, 00165 Rome, Italy; marco.spada@opbg.net; 4Hepatology, Gastroenterology, Nutrition, Digestive Endoscopy and Liver Transplantation Unit, Bambino Gesù Children’s Hospital, IRCCS, 00165 Rome, Italy; giuseppe.maggiore@opbg.net (G.M.); andrea.pietrobattista@opbg.net (A.P.); 5O.U: Radiology, Bambino Gesù Children’s Hospital, IRCCS, 00165 Rome, Italy; lidia.monti@opbg.net; 6Pediatric Hematology/Oncology, Bambino Gesù Children’s Hospital, IRCCS, 00165 Rome, Italy; aurora.castellano@opbg.net; 7O.U. Pathology, Fondazione Policlinico Universitario Agostino Gemelli, IRCCS, 00168 Rome, Italy; mariacristina.giustiniani@policlinicogemelli.it; 8Department of Medical-Surgical Sciences and Biotechnologies, Sapienza University of Rome, Polo Pontino, 00185 Rome, Italy

**Keywords:** FALD, hepatocellular carcinoma, cholangiocarcinoma, NGS, *FGFR* genes, *GNAS*, TMB

## Abstract

**Simple Summary:**

Fontan-associated liver disease (FALD) is a late sequela of single ventricle palliation. The long-term consequences of Fontan’s physiology have a spectrum of outcomes resulting in chronic hepatic venous congestion, liver cirrhosis, hyperplastic nodules, and liver neoplasms that are both benign, such as hepatic adenoma (HA), and malignant, including hepatocellular carcinoma (HCC) and cholangiocarcinoma (CC). To date, genetic and molecular studies of HCC/CC contextual to FALD are lacking. We report the molecular profile of the heterogeneous group of tumors observed in the FALDs (HA, HCC, and CC) we studied with the aim of comparing any differences with the same tumors without FALD and identifying possible recurrent variables, such as molecular alterations, that could be predictive of clinical outcome.

**Abstract:**

Purpose: Univentricular heart is corrected with the Fontan procedure (FP). In the long term, so-called Fontan-associated liver diseases (FALDs) can develop. The aim of this study is to analyze the molecular profile of FALDs. Methods: FALDs between January 1990 and December 2022 were reviewed for histology and immunohistochemistry, laboratory data, and images. Targeted next generation sequencing (NGS), performed on the DNA and RNA of both neoplastic and non-lesional liver tissue, was applied. Results: A total of 31/208 nodules > 1 cm in diameter were identified on imaging, but a liver biopsy was available for five patient demonstrating the following: one hepatocellular adenoma (HA), two hepatocellular carcinomas (HCCs), one fibrolamellar carcinoma (FLC), and one intrahepatic cholangiocarcinoma (ICC). Molecular analysis showed a copy number alteration involving *FGFR3* in three cases (two HCCs and one ICC) as well as one HCC with a hotspot mutation on the *CTNNB1* and *NRAS* genes. Tumor mutational burden ranged from low to intermediate. A variant of uncertain significance in *GNAS* was present in two HCCs and in one ICC. The same molecular profile was observed in a non-lesional liver. A *DNAJB1-PRKACA* fusion was detected only in one FLC. Conclusions: Neoplastic FALDs show some unusual molecular profiles compared with non-Fontan ones. The presence of the same alterations in non-lesional cardiac cirrhosis could contribute to the development of FALD.

## 1. Introduction

Fontan procedure (FP) is the standard surgery used on a congenital univentricular heart. The new circulation is characterized by single ventricle circulation, non-pulsatile pulmonary as well as inferior vena cava (IVC) perfusion, systemic venous hypertension, and myocardial fibrotic replacement [1]. Patients with FP have prolonged survival into adulthood with improved quality of life. However, the new circulation may develop certain dysfunctional side effects involving several organs (kidney, bowel, and liver) at an unpredictable time. In particular, the progressive narrowing of the conduit causes the elevation of the pressure in the IVC, thereby impeding the egress of the blood flow from the liver. As a consequence, non-pulsatile pressure transmits directly from the IVC to the hepatic veins (HVs) and hepatic sinusoids. Due to elevated pressure within the HVs and sinusoids, the portal venous inflow is compromised [2]. A concomitant depressed cardiac output may participate in the reduction of the portal inflow from the splanchnic circulation. When the portal blood flow decreases, the hepatic artery is dilated in a complementary way to constantly maintain hepatic blood flow and oxygen supply to the liver [3] (Figure 1).

Thus, after FP, liver parenchyma may be characterized by dilation and congestion of the hepatic sinusoids. The hampered outflow resulting from thrombus formation and hepatocyte hypoxia lead to the development of hypernodular lesions in hypoxic areas, particularly in areas periphery to the liver, due to the increased hepatic artery blood flow. Additionally, hypoxia and thrombosis within the sinusoids promote the activation of hepatic stellate cells. In congestive hepatopathy, venous congestion elicits a fibrotic response with a sinusoidal pattern. Over time, injury precipitates formation of broad scars, bridging fibrosis and cirrhosis [5]. In this context hepatic masses, encompassed under the acronym FALD (Fontan-associated liver disease), may develop. These range from focal nodular hyperplasia (FNH)-like nodules and benign tumors, such as hepatic adenomas (HAs), to malignant lesions, including hepatocellular carcinoma (HCC) and intrahepatic cholangiocarcinoma (ICC) [6]. Cirrhosis, a potential prerequisite for HCC, may appear before the age of 25 approximately 10 to 15 years after Fontan operation. Based on previous studies, the incidence of cancer per year is estimated to be about 1.5 to 5% [7,8,9].

The etiology of neoplastic lesions, in particular of malignancies, is probably multifactorial, with causes including an underlying genetic predisposition, the presence of risk factors (aging, virus infections, alcohol assumption, steatosis), and the abnormal physiology of chronically elevated central venous pressures. However, genetic and molecular studies of malignancy in the setting of FALD are lacking to date [10].

We report the genetic and molecular profile of histologically proven hepatic lesions in a monocentric series of FALDs to correlate histological features with molecular finding and to explore the biology as well as eventually potential driver genes.

## 2. Methods

### 2.1. Case Selection

Our institutional review board approved this single-institution retrospective review and waived the requirement for informed consent. Between 1 January 1990 and 31 December 2022, 450 children underwent FP for the palliative correction of a congenital univentricular heart at the Cardiac Surgery Department at Bambino Gesù Children’s Hospital. A total of 208 of them were older than 18 years of age, with an average age of 28 years and with a follow-up longer than 22 years in 2022. During the follow-up controls (through imaging and alpha-fetoprotein serology—AFP) a nodular lesion was detected in 81/208 patients. A liver biopsy was performed in suspicious nodules due to increased size and/or AFP levels, which was followed by tumor resection when indicated.

### 2.2. Clinical Features

Medical records of the patients with a nodule > 1 cm were reviewed for sex, age at FP, type of congenital heart disease (CHD), type of FP, histopathologic diagnosis of the lesion, AFP values closest prior to biopsy, and follow-up. Liver function tests, total bilirubin, and platelet count were also analyzed. Abdominal ultrasound (US) and magnetic resonance imaging (MRI) were performed in patients with nodules > 1 cm.

### 2.3. Histology

Liver biopsy or surgical sample were formalin-fixed and paraffin-embedded. Hematoxylin-eosin (HE) and Masson’s trichrome stains to assess fibrosis were routinely made. Immunohistochemistry (IHC) with anti-β-catenin (Novocastra, Buccinasco (MI), Italy NCL-L-B-CAT, 1:150 for 30 min), anti-glutamine synthetase (GS-6, Millipore, Limerick, PA, USA, MAB302, 1:400 for 25 min), anti-glypican3 (Gly3, BioMosaics, Santa Barbara, CA 93111 USA B0025R, B0055R, 1:1000 for 30 min), anti-liver fatty acid bending protein (LFABP, Abcam, Cambridge, UK ab 7807, 1:100 for 30 min), and anti-SOX9 (Cell Signaling Technology, Danvers, MA, USA, #82630, 1:100 for 30 min) were performed using Dako Omnis, Agilent Technology (Santa Clara, CA, USA).

### 2.4. DNA Analysis

DNA was extracted from formalin-fixed and paraffin-embedded (FFPE) tumor tissue and non-lesional adjacent liver tissue using Maxwell CSC instrument (Promega, Madison, WI, USA) with the Maxwell RSC DNA FFPE kit (Promega, Madison, WI, USA) according to the manufacturer’s protocols; DNA concentrations were measured on a Qubit 2.0 fluorometer (Thermofisher Scientific, Waltham, MA, USA) using the Qubit dsDNA High Sensitivity Assay Kit.

For DNA library preparation and enrichment, the TruSight™ Oncology 500 Kit (Illumina) was used following the manufacturer’s instructions. Post-enriched libraries were quantified, pooled, and sequenced on a NextSeq 550 (Illumina Inc., San Diego, CA, USA). The quality of the NextSeq 550 (Illumina) sequencing runs was assessed with the Illumina Sequencing Analysis Viewer (Illumina). NGS data were analyzed with Illumina TruSight Oncology 500 Local App v2.1 [11], and variant report files were uploaded into the Pierian Clinical Genomics Workspace cloud (Pierian DX software CGW_V6.21.1).

### 2.5. RNA Analysis

RNA was extracted from FFPE tumor tissue using Maxwell CSC instrument (Promega, Madison, WI, USA) with the Maxwell RSC RNA FFPE kit (Promega, Madison, WI, USA) according to the manufacturer’s protocol; RNA concentrations were measured on a Qubit 2.0 fluorometer (Thermofisher Scientific, Waltham, MA, USA) using Qubit RNA High Sensitivity Assay Kit, and the quality and quantity of RNA samples was ascertained through the use of the Agilent 2200 Tapestation system (Agilent Technologies). The mean RNA integrity number (RIN) was 5.2 (range from 3.0 to 6.9).

In order to identify genomic rearrangement, a targeted RNA-Seq was performed using 300 nanograms of total RNA. SureSelect^XT^ HS2 RNA system with Human All Exon V6 + COSMIC Probe (Agilent Technologies, Santa Clara, CA, USA) was used according to the manufacturer’s instructions (version A1, September 2020). The sequencing run was performed in paired-end mode (2 X 151-bp reads) using the Illumina NextSeQ 550 platform and generating at least 30 million reads per sample. The resulting alignment files were then used by the STAR-Fusion and Arriba pipelines to identify any candidate fusion transcripts [12,13].

## 3. Results

### 3.1. Cases

A surveillance program with a yearly scheduled US was undertaken. Among the 81 cases with an identifiable nodule on the US, 50/81 (62%) patients had a nodule < 1 cm in diameter; these nodules were followed up but not biopsied because AFP was low and their size did not change over time. These subcentimetric nodules found in the US examination were hyperechoic consistent with regenerative nodules. Nevertheles, an US examination of the liver was performed every six months. The remaining 31/81 (38%) patients had nodule/s > 1 cm. These cases were investigated with MRI or, in the case of general/local contraindication (noncompliance, pacemaker), with multidetector computed tomography (MDTC). In 25 cases both examinations were performed.

The features of patients with a nodule > 1 cm are reported in Table 1.

All patients showed a variable degree of disturbed architecture consistent with new circulation after FP through the ultrasound and MRI; however, only in patients who underwent a liver biopsy and resection was it possible to document congestive (cardiac) cirrhosis.

Portal inflammation and fibrosis, steatosis, apoptotic/Mallory bodies, ceroid-laden macrophages, and iron deposition, generally observed in viral/alcoholic cirrhosis, were not hallmarks of our FALDs. An increase in serum gamma-glutamyltransferase was the most common and early laboratory liver abnormality, while aminotransferases were within normal range. No patients developed esophageal varices. AFP was low in 25/31 and high in 6/31; through imaging (both abdominal ultrasound and MRI) these nodules were consistent with macroregenerative/benign hepatocellular hyperplastic nodules in 17 patients, FNH in 2, dysplastic nodule in 2, and suspicious for a tumor in 10 (6 with a high level of AFP). In the last group, 3/10 patients with an imaging and serological diagnosis of HCC were treated with transarterial chemoembolization (TACE) without a histologically confirmed diagnosis; 1 patient underwent biopsy of the liver mass in another center; an ultrasound and MRI established the differential diagnosis between an adenoma and FNH in 1 patient with low AFP; and 5 patients underwent a biopsy at our center.

In Table 2, the details of all the clinical data for these five patients are reported. A total of 3/5 patients were female, with their age at the time of FP ranging between 2 and 8 years. All had an extracardiac Fontan pathway Age at FALD: the median age at the time of their liver biopsy was 26 years (range of 20–34 years) and the interval from Fontan operation to biopsy was 20 years (range of 10–28 years).

Four patients underwent surgical resection of the hepatic mass after diagnostic biopsy, while one received only a biopsy and died due to rapid tumor progression. Four out of five (80%) patients with FALDs had an elevated serum GGT (range of 94–154 unit/L). The only patient (P3) that had a normal serum GGT was the patient who had undergone cardiac transplant 20 years after the Fontan operation. All patients had normal AST and ALT values as well as Bilirubin values. Serum AFP levels were elevated in 3/5 patients (60%). A marked increase was observed in patients with conventional HCC, while serum AFP was normal in patients with FLC and ICC. The AFP level was only minimally above the upper limit of normal in the patient (P5) who had five separate nodules and HAs from the perspective of histopathology.

The imaging patterns of the five patients are reported in Table 3. Regarding the four biopsy-proven malign nodules, the US did not result in efficacious diagnoses since there were two cases with a hypo-echogenic pattern and two with an iso-echogenic pattern. We should note that upon histology, there were three HCC cases and one ICC case. The MDCT and MRI showed a hyperdense pattern with wash-in at the arterial phase in the two HCC cases and in the fibro-lamellar case. Wash-out in the portal phase was observed in all the three cases (Figure 2a,b). Late ring enhancement was observed in all the HCC cases. The patient with ICC was only investigated with MDCT. Although it resulted in a hyperdense pattern in the arterial phase and hypodense pattern in the portal phase, no late ring enhancement was observed. The benign nodule (P5, adenoma) showed imaging features that were not diagnostic. The diagnosis was achieved through histopathology (Figure 2c,d).

One patient had multiple lesions targeted for biopsy, with a total number of seven nodules examined. Two patients with biopsied liver lesions and subsequently underwent a heart transplantation (HTx) one month and two years later, respectively. In addition, one of these patients also developed a subsequent lesion biopsied five years after HTx, likely due to the persistent influence of Fontan circulation even after HTx [6].

P1 had a recurrence of FLC two years after the resection; P2 died six months after the resection due to the rapid recurrence and progression of HCC; P3 is alive without any recurrence; P4 died one year after the histological diagnosis of HCC due to the progression of an intractable tumor; and P5 is alive and he presented other small nodules with low AFP during the follow up. He presented the germline *HNF1A* mutation, so these nodules are consistent with those that are *HNF1A*-HA-related.

Background liver (non-lesional) tissue was available in all but one case (P4) and was analyzed in pairs with neoplastic tissue.

### 3.2. Histological Features

Biopsy of background (non-lesional) liver tissue showed changes consistent with Fontan pathophysiology in all cases, including sinusoidal dilatation with portal and pericentral fibrosis (Figure 3a). One lesion was a hepatocellular adenoma (HA) belonging to the hepatocyte nuclear factor 1α [HNF1α] inactivated subtype (Figure 3b). Four malignancies included a fibrolamellar carcinoma (FLC) (Figure 3c), two trabecular HCC (Figure 3d), and one included an intrahepatic cholangiocarcinoma (ICC, Figure 3e), with an adjacent area comprising a focal nodular hyperplasia (Figure 3f) in a patient with a previous regenerative nodular hyperplasia (seven years before). Histologic features of HCCs and ICCs were indistinguishable from those occurring outside the context of FALD. Two HCCs were moderately differentiated carcinomas with a classic trabecular pattern, and one was a mixed tubular/acinar/solid ICC. FLC was composed of large polygonal cells with abundant eosinophilic cytoplasms, large vesicular nuclei, and large nucleoli in association with lamellar bands of fibrosis. An unusual morphologic appearance was represented by nests of neoplastic hepatocytes or hepatocytes in a lace-like pattern embedded in an abundant background of a extracellular myxoid matrix.

Immunohistochemically, β-catenin was negative for nuclear accumulation in all cases. Glypican3 and GS were negative in HAs and in FLC but diffusely positive in both trabecular HCCs. SOX9 was multifocally positive only in trabecular HCC. The neoplastic cells in HNF1a-HA, as in FLC, were negative for LFABP, while its expression was preserved in inflammatory HAs and trabecular HCCs.

### 3.3. Molecular Features

A total of 19 single nucleotide variants (SNVs) and three copy number alterations (CNAs) from 500 genes were identified in five samples. The most commonly altered genes in our cohort were *FGFRs*; 1/1 ICC (P3) and 2/3 HCCs (P1 and P2) showed the loss of one copy of the *FGFR3* gene, and one of these HCCs (P2) additionally showed an *FGFR2* variant. Furthermore, patients (P1 to P2) showed a variant on *GNAS*. One HCC (P4) showed a hotspot mutation on *CTNNB1* and *NRAS*; the only ICC (P3) displayed a variant of unknown significance (VUS) on *CTNNB1*.

No variants in either the *TERT* promoter or *TP53* were identified in the HCCs as well as how no *TP53*, *CDKN2A/B*, *KRAS*, and *IDH1* variations and no *FGFR2* fusion were present in the ICC.

TMB was usually low (TMB < 5), and only one case showed an intermediate TMB (TMB ranging from 5 to 10).

The non-lesional liver displayed a similar genomic profile of tumor tissue in two cases (P2 and 3), suggesting that the underlying molecular profile (congestive cirrhosis) may be common in cases of synchronous/metachronous lesions. Only one case (P1) had a non-lesional tissue profile without CNAs or variants. DNA from non-lesional tissue in P4 and P5 was not available.

RNA sequencing, performed in the four cases of P1 to P3 and P5 (one FLC, one HCC, one ICC, and one HA), identified a *DNAJB1-PRKACA* in-frame fusion in FLC. RNA was not available in the P4 case. One NRH and the inflammatory HA were not included in the molecular study. Table 4 summarizes the molecular results.

## 4. Discussion

Liver tumors present various risk factors; the main culprits are genetic/molecular alteration (hepatic adenoma), chronic viral infection, excessive alcohol intake (HCC), and primary sclerosing cholangitis (CC). These conditions often trigger the sequence of chronic inflammation—fibrosis—cirrhosis, ultimately leading to carcinogenesis. On the other hand, in vascular liver diseases, the development of liver tumors is unique and linked to rheological, vascular, genetic, and metabolic factors. Complex liver involvement is common following the Fontan procedure, from passive venous congestion to hepatic ischemia and chronic congestive hepatopathy (cardiac cirrhosis). The higher frequency of malignant nodules is less understood, and the discovered liver nodules in FALD give rise to other challenges than when they are found in the more classical clinical context.

In the current series, FALD occurred in 81/208 patients (39%), with a mean time between Fontan procedure and liver disease being 28 years. In 50/81 (62%) nodules were <1 cm, while in 31/81 (38%) the nodules were >1 cm. A biopsy was performed only in five patients with a histologic diagnosis of HA (1), FLC (1), HCC (2) and ICC (1). These data confirm the heterogeneity of malignancies arising in FALD, although all occurred in the context of a cirrhotic background. In all HCC and ICC cases, immunostains for β-catenin were negative (absence of nuclear accumulation); GS, Glypican3, and SOX9 were negative in FLC and focal or diffuse positive in trabecular HCC. *CTNNB1* point mutation in exon 3 was present in one trabecular HCC, which was in line with the reported mutation rate of 35% in HCC outside the context of FALD [14]. *FGFR3* allele loss was detected in two HCCs as well as in one ICC, although this loss did not lead to the lack of FGFR3 expression. The only identification in *FGFR3*’s copy number alteration did not exclude the possibility of a gross loss of a chromosomal region containing important genes. No variants in either the *TERT* promoter or *TP53* were identified in the HCCs, and no *TP53*, *CDKN2A/B*, *KRAS*, *ARID1A*, and *IDH1* variations or *FGFR2* fusion were present in the ICC.

Fibroblast growth factors (FGFs) in the liver have been shown to promote regeneration. Fibroblast growth factors bind distinct receptors that mediate different effects. Overall, the overexpression of FGF receptors (FGFRs) seems to drive HCC development and progression in classic contexts [15]. *FGFR3* amplification has been reported in HCC and plays a significant role in HCC development and progression, and FGFR3 expression is elevated in human cirrhotic livers. In contrast, in the current series of FALDs, one allele loss in three cases (three HCC and one ICC) is an intriguing finding and deserves further investigation. Although RNA-seq demonstrated a reduction of *FGFR3* expression in one case, it is difficult to know if the loss of one allele plays a role in pathogenesis. However, the occurrence of this alteration in three cases may act as a predisposition to tumor development, also being present in non-tumor tissues.

The evidence of *FGFR3* alteration in non-tumor tissue is particularly intriguing. In fact, while the functional role of FGFR3 and its isoforms has not yet been investigated in the context of hepatic fibrosis, its cytoprotective function in hepatocytes has been demonstrated in experimental models. Mice lacking FGFR3 show increased liver necrosis, particularly in response to CCl4 treatment, with enhanced fibrosis [16]. Furthermore, FGFR3 in hepatocytes can directly limit fibrosis by suppressing the expression of pro-fibrotic molecules, such as Loxl4 and Tff3, that are expressed at higher levels in the hepatocytes of Alb-R3 mice [16]. Thus, we can speculate that the loss of *FGFR3* may be involved in the development of cirrhosis rather than in neoplastic transformation in our cases. FGFR3 variants need further investigation; however, they might be related to the microenvironment generated by FP. Nonetheless, it is interesting to note that mutations in FGFR3 and FGFR4, pluripotency signaling, and developmental genes have been identified in hepatocellular neoplasms that have not otherwise specified (HCN NOS) particularly challenging liver tumors with combined or overlapping histological features of hepatoblastoma and hepatocellular carcinoma observed in late pediatric age [17].

Another common alteration in our series comprises *GNAS* variants. The *GNAS* gene encodes the alpha subunit of the stimulatory G protein, which regulates neurotransmitters and many hormones through generating cAMP. *GNAS* mutations are highly associated with McCune–Albright syndrome; however, *GNAS*-activating mutations have also been reported in a subgroup of inflammatory liver adenomas and rarely in HCCs with the activation of a signal transducer and of an activator of transcription 3 C (STAT3) [18], mostly with a fibrotic pattern [19]. In our series, the *GNAS* variants identified were of VUS, so the significance is unknown and the possible association with cardiac cirrhosis remains to be explored.

A particularly interesting finding in the current series is the occurrence of an FLC case that presented the pathognomonic fusion transcript *DNAJB1-PRKACA*. This alteration was absent in the non-lesional adjacent liver tissue. To the best of our knowledge, this is the second FLC reported among FALDs [20]. Although it may be a casual association, the rarity of the histotype and the unusual pattern are intriguing.

HCCs have a generally low (<5 muts/Mb) TMB and are very rarely hypermutated [21]. A tumor mutation burden (TMB) is an emerging biomarker that is also predictive of the response to immune checkpoint inhibitors, and patients with a higher TMB are more likely to obtain benefits from immunotherapy. Measuring TMB (as total number of somatic nonsynonymous mutations present in the coding region expressed as mutations per megabase in a tumor—muts/Mb) [22] is currently graded as being low, intermediate, or high. Many genomic alterations have been found to be associated with different TMBs, though there is no single oncogenic driver mutation in the majority of HCCs. In our cohort, one trabecular HCC presented an intermediate (7.8/Mb) TMB, and the patient presented a rapid progression of the tumor and died one year after the diagnosis.

Routine screening for liver disease in Fontan patients remains a challenge. Noninvasive measurements of the hepatic function remain intact until advanced stages of hepatic fibrosis. In order to determine which lesions are potentially benign, identifying reactive lesions versus malignant ones is one of the difficulties in FALD evaluation. Liver tumors develop approximately 10 to 20 years after FP, frequently with a cirrhotic background and with elevated AFP. It is important to underline that Fontan liver physiology is primarily a congestive condition; therefore, HCC/ICC in this situation could have peculiar histologic, immunohistochemical, and molecular features and cannot be treated using the same protocols as post-viral infection, alcohol abuse, or NASH-related HCCs [4].

The variety of FALDs reported in both the literature and our cohort is similar to what has been reported in the presence of congenital vascular anomalies of the liver, such as Abernethy malformations, Budd–Chiari syndrome, or congenital porto-systemic shunts. The abnormal hepatic vascularity (acquired or congenital) generates diffuse abnormal chronic passive hepatic congestion, centrilobular necrosis, and cardiac cirrhosis that sometimes evolves into liver lesions of multiple histologic subtypes. The specific processes that contribute to lesion development in patients following the FP as well as other hepatic vascular abnormalities are incompletely defined. Hepatocellular nodules arising in the context of FALD as well as other vascular liver diseases are remarkably diverse, ranging from regenerative to benign to malignant hepatocellular tumors [23]. Knowledge of the molecular profile of these lesions, in particular of malignant tumors, could help shed light on their pathogenesis.

Liver biopsy is the gold standard for diagnosing and staging most adult liver diseases. Nevertheless, in the context of FALDs, a biopsy has several limitations including sampling error, risk of bleeding and infection, and the need for procedural sedation; therefore, a liver biopsy is at least highly recommended in cases of suspicious imaging (i.e., increase in size, focality, and growth of the nodule as well as the presence of rings or behaviors with wash-in and wash-out in the MRI phases) or high levels of AFP. Hepatic samples are useful both for the definitive diagnosis and for exploring the molecular profile of tumors. Finally, as recently suggested by an EASL-ERN position paper, implementation of screening programs may help to identify at-risk candidate populations [24], and the analysis of the molecular features of tumors in FALDs may be an important piece of the puzzle.

## 5. Conclusions

The prevalence of FALDs is increasing worldwide, and the frequency of liver complications is rising because of the improvement of cardiac survival. Despite the limited number of cases being insufficient for defining a distinctive molecular profile of the tumors in FALDs, our series demonstrates their morphologic peculiarities and unique characteristics, especially in malignancies. Combining these features with the specific type of tumor will provide unprecedented insight into how the tumor forms and identify the mechanisms and pathways regulating hepatocyte/cholangiocyte neoplastic differentiation in livers post-FP. This insight will not only address questions of fundamental biological significance but will provide insight into the causes of tumorigenesis, representing a blueprint for generating biologically relevant in vitro models and aiding in the development of cell-based regenerative strategies to treat disease. Multi-center prospective studies and more in-depth investigations into the pathophysiology of FALDs are desirable.

## Figures and Tables

**Figure 1 cancers-16-00307-f001:**
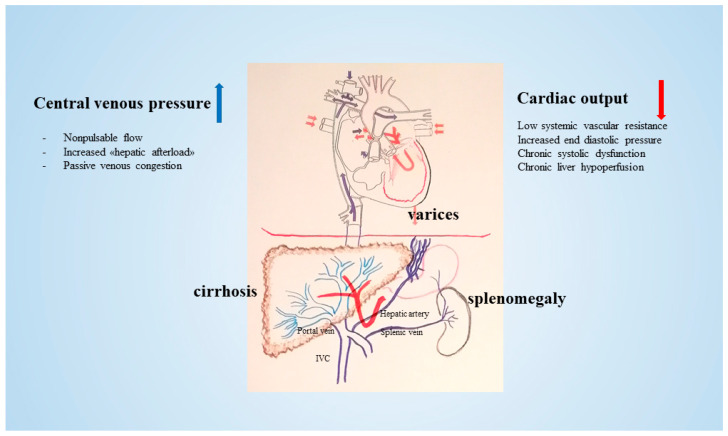
After FP, increased central venous pressure and non-pulsatile flow cause passive venous congestion secondary to increased hepatic afterload. Decreased cardiac output and increased central venous pressure ultimately lead to cirrhosis and subsequently to varices, splenomegaly, and portal hypertension [4].

**Figure 2 cancers-16-00307-f002:**
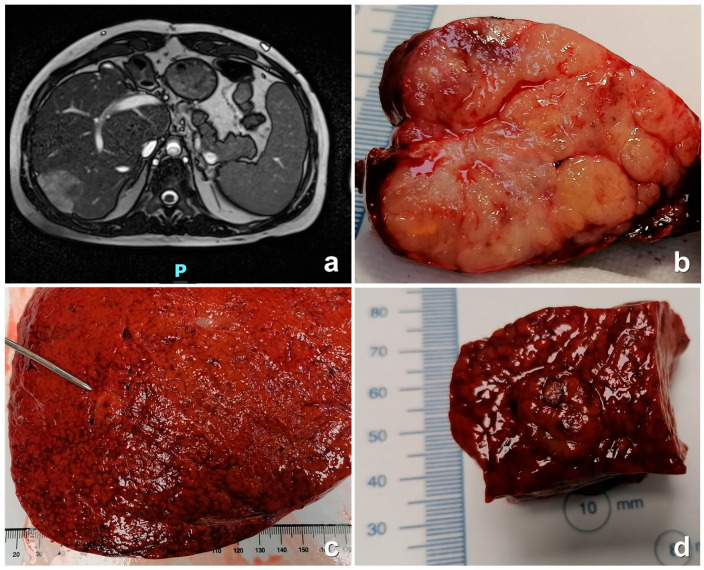
RMI: Diffusely heterogeneous macronodular liver with irregular margins. In the VII hepatic segment (in the subglissonian site) there is a nodular formation (with blurred edges) that is inhomogeneous, with maximum axial dimensions of approximately 4.2 cm (AP) × 4.5 cm (LL) × 4 cm (CC) (**a**). Surgical specimen: tumor appears well demarcated and has a nodular appearance (**b**). Liver resection: the parenchyma is congested and has irregular architecture (**c**). The probe shows a round mass (**d**). P = posterior.

**Figure 3 cancers-16-00307-f003:**
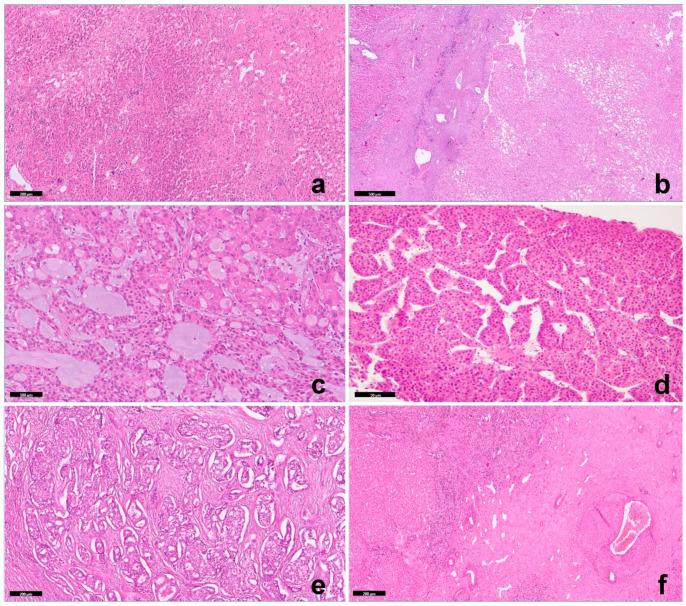
(**a**) Non-lesional liver: sinusoidal dilatation, HE, 5×; (**b**) Hepatic adenoma that is HNF1a-linked: portal tract absence and steatosis, and non-lesional liver is on the left, HE, 2×; (**c**) FLC with unusual one myxoid area: neoplastic cells are embedded into a myxoid background, HE, 10×; (**d**) Trabecular hepatocellular carcinoma: moderately differentiated clear cells of HCC, HE, 40×; (**e**) Intrahepatic cholangiocarcinoma: adenocarcinoma with variable morphological aspects of tubular structures and acini formation formed by columnar to cuboidal epithelial cells, which resemble biliary epithelial cells, HE, 5×; (**f**) Focal nodular hyperplasia: central scar with arterial wall remodeling from intimal hyperplasia, HE 2×.

**Table 1 cancers-16-00307-t001:** Demographic median results from laboratory tests and abdominal ultrasounds for all patients with a nodule > 1 cm.

Parameter	Number
N° patients	31
Gender F/M	17F 14M
Age at Fontan (years)	2–22 years
CHD	
TA	10
Univentricular	6
PA with intact IS	4
ccTGA	4
AVSD	3
HLHS	2
DORV	1
DILV	1
Fontan type	
Extracardiac	28
Intracardiac	3
Laboratory test	
ALT average (range)	37.5 (14–61)
AST average (range)	31.5 (14–47)
GGT average (range)	127 (22–232)
Bil tot average (range)	1.41 (0.35–2.48)
AFP (range)	0.9–11,478
Abdominal ultrasound	
Hepatomegaly Y/N	16/13 *
Splenomegaly Y/N	12/17 *

**Abbreviations**: TA = tricuspid atresia; PA with IS = pulmonary atresia with intact interventricular septum; ccTGA = congenital correct transposition of great arteries; AVSD = atrio-ventricular septal defect; RV = right ventricle; HLHS = hypoplastic left heart syndrome; DORV = double outlet right ventricle; and DILV = double-inlet left ventricle. * two cases lost in the follow up.

**Table 2 cancers-16-00307-t002:** Demographic median results from laboratory tests and prognosis for all patients with nodule biopsy or tumor resection.

Patient	M/F	Age at FP (y)	CHD	Age at Diagnosis (y)	Interval Fontan-FALD (y)	GGT n.v. 11–65	Platelet n.v. 150–450	Serum AFP n.v. < 8.10	Tumor Size	Prognosis/Death
P1	F	4	Complete AVSD—RV dominant	20	16	100	208 × 10^3^/uL	2.43 ng/mL	4.5 cm	Alive HTx 2019
P2	F	3	TA-PA	22	22	99	175 × 10^3^/uL	11,458.00 ng/mL	4.5 cm	† (6 m post-FALD)
P3	M	2	DILV and TGA	18 22	16 * 20 **	37	173 × 10^3^/uL	** 447.8 ng/mL	8 cm	Alive HTx 2014
P4	F	6	TA-PSt-TGA	34	28	154	182 × 10^3^/uL	3005.00 ng/mL	3 cm	† (1 y post-FALD)
P5	M	8	PA with intact IS	32	16	94	111 × 10^3^/uL	9.43 ng/mL	2.3 cm	Alive

**Abbreviations**: AVSD = atrio-ventricular septal defect; RV = right ventricle; TA-PA = tricuspid atresia-pulmonary atresia; DILV = double-inlet left ventricle; PSt = pulmonary stenosis; PA with IS = pulmonary atresia with intact interventricular septum. TGA = transposition of great arteries. * = 1° nodule, ** = 2° nodule, † = died.

**Table 3 cancers-16-00307-t003:** US-MDCT-MRI features of all patients with a nodule biopsy or tumor resection.

	US	MDCT			MRI		
		Arterial Phase	Portal Phase	Late Phase	Arterial Phase	Portal Phase	Late Phase
P1 FLCA	Iso-echoic	Slightly hyper-dense Wash-in	Hypo-dense Wash-out	Slightly hypo-dense with hyperdense ring	Hyper-intense Wash-in	Hypo-intense Wash-out	Hypo-intense
P2 HCC	Hypo-echoic	Hyper-dense Wash-in	Hypo-dense Wash-out	Hypo-dense	Hyper-intense Wash-in	Hypo-intense Wash-out	Hypo-intense
P3 ICC	Hypo-echoic	Hypo-dense No wash-in	Hyper-dense	Hypo-dense	ND	ND	ND
P4 HCC	Iso-echoic	Slightly hyper-dense Wash-in	Hypo-dense Wash-out	Slightly hypo-dense with hyperdense ring	ND	ND	ND
P5 HA	Hyper-echoic	Slightly hyper-dense No wash-in	Hyper-dense	Hypo-dense Wash-out	Iso-intense	Hyper-intense	Hypo-intense

**Table 4 cancers-16-00307-t004:** Molecular data from lesional and non-lesional tissue.

	P1 FLC	P1 Non-Lesional Tissue	P2 HCC	P2 Non-Lesional Tissue	P3 CC	P3 Non-Lesional Tissue	P4 HCC	P5 Adenoma
DNA Analysis								
TMB	3.9	0.8	7.8	5.5	3.9	3.1	0.8	3.1
MSI	3.30%	2.50%	1.90%	0.00%	1.10%	1.90%	0.90%	5%
CNA	*FGFR3* (NM_000142.4) 1 allele loss	None	*FGFR3* (NM_000142.4) 1 allele loss	*FGFR3* (NM_000142.4) 1 allele loss	*FGFR3* (NM_000142.4) 1 allele loss	*FGFR3* (NM_000142.4) 1 allele loss	None	None
Variants	*GNAS* (NM_080425.2) c.1299A > G; p.(Gln371Arg) [VAF 3%]	wt	*FGFR2* (NM_022970.3) c.1127A > G; p.(Tyr376Cys) [VAF 6%]	*FGFR2* (NM_022970.3) c.1127A > G; p.(Tyr376Cys) [VAF 13%]	*LRP1B* (NM_018557.2) c.9482dup; p.(Asp3161GlufsTer15) [VAF 7%]	*LRP1B* (NM_018557.2) c.9482dup; p.(Asp3161GlufsTer15) [VAF 6%]	*CTNNB1* (NM_001904.3) c.110C > T; p.(Ser37Phe) [VAF: 12%]	*HNF1A* (NM_000545.5) c.489C > A; p.(Tyr163Ter) [VAF 39%)
	*RASA1* (NM_002890.2) c.2708G > C; p.(Arg903Pro) [VAF 11%]	wt	*GNAS* (NM_080425.2) c.1531C > T; p.(Arg511Cys) [VAF 57%]	*GNAS* (NM_080425.2) c.1531C > T; p.(Arg511Cys) [VAF 57%]	*CTNNB1* (NM_001904.3) c.569G > A; p.(Arg190His) [VAF 21%]	*CTNNB1* (NM_001904.3) c.569G > A; p.(Arg190His) [VAF 21%]	*NRAS* (NM_002524.4) c.182A > G; p.(Gln61Arg) [VAF: 12%]	*MSH3* (NM_002439.4) c.2623G > A; p.(Asp875Asn) [VAF 44%)
	*MAP2K2* (NM_030662.3) c.142G > C; p.(Glu48Lys) [VAF 19%]	wt	*MSH3* (NM_002439.4) c.302del; p.(Val101GlufsTer25) [VAF 6%]	wt	*KDM6A* (NM_021140.2) c.2689C > G; p.(Leu897Val) [VAF 99%]	*KDM6A* (NM_021140.2) c.2689C > G; p.(Leu897Val) [VAF 99%]	*NOTCH1* (NM_017617.3) c.4843A > T; p.(Met1615Leu) [VAF:48.1%] + c.3739C > T; p.(Gln1247Ter) [VAF:47%]	
			*ATRX* (NM_000489.3) c.3634_3636del; p.(Asp1212del) [VAF 37%]	*ATRX* (NM_000489.3) c.3634_3636del; p. (Asp1212del) [VAF 40%]	*GNAS* (NM_080425.2) c.1299A > G; p.(Gln371Arg) [VAF 24.9%]	*GNAS* (NM_080425.2) c.1299A > G; p.(Gln371Arg) [VAF 25%]		
			*PALB2* (NM_024675.3) c.3103A > G; p.(Ile1035Val) [VAF 43%]	*PALB2* (NM_024675.3) c.3103A > G; p.(Ile1035Val) [VAF 52%]	*RANBP2* (NM_006267.4) c.7216G > A; p.(Ala2406Thr) [VAF 10%]	*RANBP2* (NM_006267.4) c.7216G > A; p.(Ala2406Thr) [VAF 10%]		
RNA Analysis								
Fusion genes	*DNAJB1-PRKACA*	Negative	Negative	Not analyzed	Negative	Not analyzed	Not available	Negative

**Abbreviations**: TMB = tumor mutational burden; CAN = copy number alteration; MSI = microsatellite instability; wt = wild type; VAF = variant allele frequency.

## Data Availability

Data supporting reported results have been found in archived datasets analyzed and generated during the study.

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
