# Peer review of "Histopathological Spectrum and Molecular Characterization of Liver Tumors in the Setting of Fontan-Associated Liver Disease"

_cancers, 2024, doi:10.3390/cancers16020307_

Round 1

Reviewer 1 Report (Previous Reviewer 2)

Comments and Suggestions for Authors

Now, the authors took into account almost all the comments and formulated their conclusions more correctly. The article can be accepted for publication. However, I would recommend that the authors add captions for the tables. Some tables still don't have captions at all. And caption for the Table 2 (it should be Table 3) "Molecular data" is too short and should be extended with details.

Author Response

We have added captions for tables.

Reviewer 2 Report (Previous Reviewer 4)

Comments and Suggestions for Authors

This is a very nice paper where the authors address liver tumors in a vascular condition.

Vascular diseases are usually associated with a variety of liver lesions and may prove a significant challenge for diagnosis.

The content is solid and references are updated. The discussion has a good correlation with other abnormalities such as Abernethy.

Two minor points for manuscript improvement:

1. An image with the physiopathology of the vascular condition and the development of liver lesions would be very interesting and put a visual in the reader's mind

2. Although there is a correlation between lever lesions in other vascular abnormalities, I would like also to see a correlation between the molecular profiles. I only saw clinical features

Author Response

We have added a cartoon with the physiopathology after FP

The genetic alterations identified in our cohort are not reported to be involved in vascular lesions.

This manuscript is a resubmission of an earlier submission. The following is a list of the peer review reports and author responses from that submission.

Round 1

Reviewer 1 Report

Comments and Suggestions for Authors

Francalanci et al. reported molecular profiles of liver tumors in the setting of Fonta-associated liver disease. This is a fascinating topic for hepatologists and surgeons.

Although this study aimed to find possible biomarkers for the disease, they only described the result of protein staining and genomic analysis. Noninvasive serum markers should be studied or mentioned. Moreover, some imaging results correspond to gene alteration, such as EOB-MRI findings to CTNNB1. Sonographic findings and dynamic CT or MRI findings would attract more readers. Fibrosis stage, Child-Pugh class, and platelet counts could also be added to the baseline characteristics.

Since this is a small case series study, more detailed information would be added. For instance, please consider adding a gross appearance or lower magnification of the liver nodule in Figure 1.

The authors mentioned there was a patient with multiple nodules. Please include the synchronous or metachronous analysis of the tumor, if possible. The relationship between genetic alteration and multicentric occurrence or intrahepatic metastasis is of great interest.

Please discuss more about why the TERT promoter mutation was not seen in the study. Can it be a specific finding to FALD-HCC?

Author Response

Thank you for review our paper

  1. We added liver function tests, total bilirubin and platelet count in Tab1. We did not mention them because they were normal or minimally abnormal and were neither contributory nor predictive of any type of FALD. The Child-Pugh classification of the severity of liver disease based on the degree of ascites, serum bilirubin and albumin concentrations, prothrombin time, and degree of encephalopathy is not currently applicable in post-Fontan patients because even if congestive (cardiac) cirrhosis is reported in approximately 60% of patients 30 years after Fontan physiology (Rathgeber sl, J Am Heart Assoc 2020, 9:e012529), only 1.5-5% of patients develop FALD, especially malignant neoplasms . Therefore, these parameters are not useful for predicting who is most at risk.
  2. We have added a figure (Figure 1) including MRI and the corresponding mass during surgery.
  1. P3 had synchronous CHF and FNH and an NRH 6 years prior (bioptized). We studied the ICC and non-injured tissue. Both showed the same molecular profile. The ICC did not show the recurrent mutations. These findings seem to suggest that the underlying molecular profile (congestive cirrhosis) may be common to all lesions.
  2. As mentioned in the Discussion “No variants of the TERT promoter or TP53 have been identified in HCCs”. This observation requires more cases, but if confirmed, would be consistent with a “FALD-HCC-specific finding. Foe sake of completeness, we added the information in the Results.

Reviewer 2 Report

Comments and Suggestions for Authors

In the manuscript “Molecular profile of liver tumors in the setting of Fontan-associated liver disease” the authors described genetic and molecular profile of liver nodules in the context of Fontan-associated liver disease (FALD). Patients with FALD have abnormal physiology and higher incidence of cancer. Although studying of such cases is very important and can give us new information about FALD and cancer development mechanisms, this study has limitations that cannot be eliminated, but also cannot be ignored.

The authors tried to study genetic and molecular profile of too low number of patients (9). Moreover, the cohort was heterogeneous: 1 regenerative nodular hyperplasia, 2 hepatic adenomas and 4 hepatocellular carcinoma, 1 fibrolamellar carcinoma and 1 intrahepatic cholangiocarcinoma. Therefore, the results obtained will not be useful either in the context of FALD or liver lesions.

For future submissions, I would recommend authors to consider the following additional concerns.

1. It would be helpful to describe in the introduction any hypothesis which molecular mechanisms the authors suggest for FALD-associated liver cancer and which they tried to study.

2. Tables should include detailed description.

3. The authors should describe the process of data analysis in more detail. The phrase that “NGS data was analyzed with Illumina TruSight Oncology 500 Local App v2.1” does not have sense because it cannot be reproduced by readers. After all steps of analysis, how did the authors select variants which are listed in Table 2?

4. All variants should be named in accordance with HGVS.

Comments on the Quality of English Language

Minor editing of English language required

Author Response

Thank you for review our paper

  1. We agree with the Reviewer that the current series is heterogeneous, it reflects also the heterogeneity of neoplasms occurring in FALD and we tried to explore the possibility of a common “denominator” represented by the pathogenetic mechanism promoting tumor development. Unfortunately, being a monoinstitutional study, it suffers from limited number of cases. However, it is important even for future studies the factthat we found “unexpected” histotype like FLC. As required we added in the Introduction a part focusing on the pathogenesis of FALD: (apparently) the driver is the new circulation (after Fontan surgery) which causes hepatic congestion and fibrosis predisposing to the development of nodules. We therefore asked ourselves why, in the same context, one patient develops FNH and others HA or CC or HCC? The first step was to analyze the molecular profile of the lesions and compare it with the same lesions in non-Fontan patients.
  2. Tables are implemented
  3. SNPs and likely benign/benign variants were excluded from the report. Only variants likely pathogenetic/pathogenetic and VUS were reported.

  1. All variants are named in accordance with HGVS

Reviewer 3 Report

Comments and Suggestions for Authors

The manuscript entitled " The manuscript entitled " Molecular profile of liver tumors in the setting of Fontan-associated liver disease " has been reviewed.

This is a very important paper. However, there are problems with the way the paper is organized: there are only five cases out of 208, which is a small number. There seems to be a limitation in drawing conclusions from the gene profiles. Therefore, it would be better to prepare the paper as a case report of five cases.It is recommended that the paper be re-written and that it should be a case report of five cases.

Others:

Is hepatocellular carcinoma developing in Fontan-associated liver disease a cirrhosis where central venous pressure is a problem?

Or are they similar to hepatocellular carcinomas associated with inflammation, such as HBV and HCV?

Cholangiocarcinoma and hepatocellular carcinoma have different mechanisms of pathogenesis; each is included in the five cases, but are they really associated with Fontan-associated liver disease?

Comments on the Quality of English Language

none

Author Response

Thank you for review our paper

  1. We agree the number of cases with documented neoplasm is low (see also answer1 to Reviewer 1) and we tried to reorganize the paper putting emphasis on description of the case series as requested. In the Introduction we implemented the pathogenesis of FALD: (apparently) the driver is the new circulation (after Fontan surgery) which causes hepatic congestion and fibrosis predisposing to the development of nodules. Congestive (cardiac) cirrhosis is a completely different background compared to cirrhosis secondary to viral or alcoholic hepatitis. The aim of our study was to report the molecular profile of neoplastic FALD (HCC and CC in particular) and the possible relevant differences with lesions arising outside this context according to existing data in literature. The heterogeneity of the lesions, including ICC and FLC is, in our opinion, intriguing and not simply-casual being CC reported in literature (Wang d et al Pathology-research and Practice 214 (2018) 914-918). In the same way, FLC (only one previous case reported) might highlight the association already demonstrated in other malignancies driven by recurrent molecular alteration in the context of treatments (e.g. other neoplasms) causing cell-stress and hypoxic damage. However, the limited number of cases prevents us from drawing conclusions.

Reviewer 4 Report

Comments and Suggestions for Authors

The authors present a very interesting paper (at least for people with an interest in liver research).

The cohort is rather small, but that is due to the rare condition under studio, and even smaller when associated with the condition of having a liver lesion and tissue sampling.

The rationale is strong and the methods are solid.

The literature seems adequate and the language is ok.

The only minor change that I suggest is to compare the genetic findings in this setting with genetic findings in other vascular malformation-associated liver tumors such as the Abernethy malformation

https://pubmed.ncbi.nlm.nih.gov/27038679/ (an example)

Author Response

Thank you for review our paper

We have had only 1 HCC in Abernethy malformation and we studied the molecular profile of this tumor. We did not find similarities with HCC in Fontan. However, only 1 case is not enough to compare. We did not include the case in the series to avoid confusion for the reader.

Round 2

Reviewer 1 Report

Comments and Suggestions for Authors

The authors have modified the manuscript as requested. I have no further comments.

Reviewer 2 Report

Comments and Suggestions for Authors

Unfortunately, the authors did not take into account all concerns, and the manuscript cannot be accepted in the present form because it is still devoid of any logical idea that the authors wanted to test. The authors described the physiological and histological changes caused by the Fontan operation. However, it is still unclear how the described changes can be associated with any genetic or molecular profiles? If the authors wanted to explore potential driver genes and the major dysregulated signaling pathways as they wrote in the end of the Introduction, they should have carried out gene expression analysis instead of searching somatic mutations. Gene expression analysis would be more logical step because all the described physiological and histological changes would influence it. However, it is unclear why did the authors expect that any specific somatic mutations should have occurred as a consequence of these changes?

The aims and conclusions did not correspond to each other:

– Simple Summary: “with the aim of identifying within this subgroup recurrent variables, such as genetic alterations or gene expression, which could be predictive of clinical outcomes such as treatment response, overall survival, and recurrence rate.” How did the authors expect to find recurrent genetic alterations, if each subgroup included only 1-2 patients? The authors did not describe gene expression, and they did not analyze clinical outcomes, treatment response, overall survival, and recurrent rate.

– Abstract: “with the aim to better understand underlying pathogenetic mechanism and the potential therapeutic targets”. What new information about pathogenetic mechanism and potential therapeutic targets did the authors find?

– Conclusion: “Despite the limited number of cases is insufficient to define a distinctive molecular profile of the tumors in FALD, our series demonstrates the morphologic peculiarity and unique characteristics, especially in malignancies”. “Molecular profile” is in the title of the manuscript and in main aims, and this phrase negates the significance of all the results obtained.

How many patients did the study include? In the abstract, “The study included 9 patients who developed FALD”. However, in the results, there are only 5 patients.

Does the phrase “The non-lesional liver displayed similar genomic profile of tumor tissue in two cases” mean that CNVs and point mutations identified were not driving for the hepatic tumors?

In the Abstract the authors wrote copy number alteration, but in the Results section, they call them as copy number variations. Were they alterations (somatic) or variations (germline)?

Tables still does not include detailed captions with all explained abbreviations.

Data analysis process is still poorly described. The authors did not take into account previous suggestions.

Also, the process of genomic rearrangements identifications should be described in more details. The phrase “In order to identify genomic rearrangement, 300 nanograms of total RNA were used for NGS library preparation with SureSelect XT HS2 kit (Agilent Technologies) following the manufacturer's instructions.” does not describe the process of library preparation. It is still unclear was it targeted RNA-seq or total RNA-seq? In the abstract, the authors wrote “targeted NGS was performed on DNA and RNA” that says about any target enrichment for RNA-seq. Was it performed?

The genomic variants are still not in HGVS format because for example “c.9482dupA” should be replaced with “c.9482dup”.

Comments on the Quality of English Language

Moderate editing of English language required

Author Response

First of all, we thank the Reviewer for his criticism. We have tried to better explain the purpose of our study. However, it is essential to consider that HCC/CC, despite being widely reported in the literature in patients undergoing the Fontan procedure, are generally described in small series, especially if coming from a single center, as in our study. This is due to the fact that the vascular substrate of these tumors is very unusual and completely different compared to the more frequent viral/alcoholic inflammatory background associated with the same tumors. Furthermore, observing the great variety of lesions, even in a small cohort like ours, we asked ourselves: why, in the same context (congestive liver disease/congestive cirrhosis), does someone develop a benign hyperplastic nodule and other benign or malignant tumors? Subsequently, we analyzed the molecular profile of the available tissue lesion by comparing it with the molecular profile (reported in the literature) of the same tumor in a patient without Fontan. We took the first step of a very complex study, knowing full well that the results would not be definitive and that a larger group would be essential.

- we examined the clinical and imaging data of 31 patients with a liver nodule larger than 1 cm and compared these data with the same tumor in NON-Fontan patients. Although histology and IHC are almost overlap, the molecular profile showed differences compared to non-FALD. Furthermore, it may be possible that the cirrhotic background, with a specific pattern of fibrosis, also has a pivotal role in carcinogenesis in FALD.

- We have changed the message in the Abstract

- We have changed the Title

- Tumor genesis in FALD likely is multifactorial, So CNA and point mutations may act together with fibrosis in triggering cell transformation.

- We analyzed only tissue samples and we cannot distinguish germline variations to somatic alteration. We can speculate that somatic alterations have an allele frequency minus 50%.

 - Tables are completed by abbreviations.

- Data analysis and molecular data have been changed following your suggestions.

We hope we have comprehensively responded to all the comments made.

Paola Francalanci

Reviewer 3 Report

Comments and Suggestions for Authors

The manuscript entitled " Molecular profile of liver tumors in the setting of Fontan-associated liver disease. " has been reviewed.

The authors have carefully restructured the paper, and this paper is stronger.